# AN EMPIRICAL STUDY ON RECONSTRUCTING SCIENTIFIC HISTORY TO FORECAST FUTURE TRENDS

## ABSTRACT

The advancement of scientific knowledge relies on synthesizing prior research to forecast future developments, a task that has become increasingly intricate. The emergence of large language models (LLMs) offers a transformative opportunity to automate and streamline this process, enabling faster and more accurate academic discovery. However, recent attempts either limit to producing surveys or focus overly on downstream tasks. To this end, we introduce a novel task that bridges two key challenges: the comprehensive synopsis of past research and the accurate prediction of emerging trends, dubbed *Dual Temporal Research Analysis*. This dual approach requires not only an understanding of historical knowledge but also the ability to predict future developments based on detected patterns. To evaluate, we present an evaluation benchmark encompassing 20 research topics and 210 key AI papers, based on the completeness of historical coverage and predictive reliability. We further draw inspirations from dual-system theory and propose a framework *HorizonAI* which utilizes a specialized temporal knowledge graph for papers, to capture and organize past research patterns (System 1), while leveraging LLMs for deeper analytical reasoning (System 2) to enhance both summarization and prediction. Our framework demonstrates a robust capacity to accurately summarize historical research trends and predict future developments, achieving significant improvements in both areas. For summarizing historical research, we achieve a 18.99% increase over AutoSurvey; for predicting future developments, we achieve a 7.71% increase over GPT-4o.

## 1 INTRODUCTION

For over 200,000 years, human intelligence has evolved, with knowledge-building processes underpinned by the dual imperatives of learning from the past and forecasting future directions (Sternberg, 2000). From the conceptual foundations of Ramon Llull's "Tree of Knowledge" to Francis Bacon's structured approach to human learning, both historical and contemporary scholars have emphasized the critical role of synthesizing past insights to drive future advancements. In recent years, modern frameworks addressing scientific discovery and knowledge structuring have further underscored this dual focus (Fire & Guestrin, 2019; Nagarajan et al., 2015).

The rapid growth of scientific publications presents an unprecedented challenge: researchers must now sift through vast amounts of literature to extract relevant historical insights and anticipate future trends (Fire & Guestrin, 2019). LLMs offer potential solutions by automating tasks such as retrieval, summarization, and analysis. However, most existing approaches either concentrate on retrospective literature reviews (Wang et al., 2024; Agarwal et al., 2024) or focus solely on generating novel research by using simple concept-level link predictions lacking semantic relationships (Krenn et al., 2023; Lu, 2021; Gu & Krenn; Tran & Xie, 2021). These narrow approaches neglect the essential integration of synthesizing past research with projecting future developments, a combination that is increasingly crucial for scientific discovery (Figure 1).

To address this gap, we propose *Dual Temporal Research Analysis* (*DTRA*), a novel task that unifies the analysis of past research with the forecasting of future trends. In contrast to traditional methodologies, which focus on either historical synthesis or future speculation, our task bridges both by leveraging past knowledge to generate informed predictions. This twofold task is especially relevant in domains such as artificial intelligence (AI), where understanding prior research trajectories is essential for predicting emerging advancements.

Figure 1: Comparison on dual temporal research analysis of a) human researchers b) current methods and c) our *HorizonAI*. Our framework resembles human researchers in the workflow while improving on thoroughness and logical reasoning, with both historical narrative and future prediction as output. In contrast, current methods focus only on either summarizing history (Wang et al., 2024; Edge et al., 2024) or generating future ideas (Baek et al., 2024; Si et al., 2024).

The *DTRA* consists of two interconnected phases: it involves consolidating and validating historical research trends, followed by the application of these insights to predict future developments. This approach mirrors the distinction between validation and experimentation, wherein past research serves as a foundation for verifying patterns and future predictions represent experimental, data-driven inferences (Chaiken, 1999; Posner, 2020).

Our framework *HorizonAI* (Figure 2) draws inspiration from Dual-System Theory (Chaiken, 1999), which posits that human cognition operates through two systems: System 1, which performs rapid, intuitive assessments, and System 2, which engages in deliberate, analytical reasoning. In the context of our framework, System 1 focuses on efficiently organizing historical data into structured formats such as temporal knowledge graphs (TKGs) (Cai et al., 2022), while System 2 conducts in-depth reasoning using Chain-of-Thought (CoT) (Wei et al., 2022; OpenAI, 2024b), to identify patterns and project future developments. Together, these systems enable a comprehensive analysis of both past and future research.

Given the novelty of the task, no established evaluation benchmarks or standardized methodologies currently exist. To address this, we introduce an evaluation benchmark *ResBench* that assesses the performance through historical completeness and predictive reliability. Extensive experiments demonstrate the superior performance of our proposed framework, *HorizonAI*, in tracing historical trends and making reliable future predictions. In comparison to existing baselines, it achieves higher predictive accuracy and generates more coherent, insightful content.

The main contributions of this paper are summarized as follows:

- **Integrating Historical Analysis and Future Forecasting**: We introduce *DTRA*, a task that combines the analysis of historical research with predictions about future trends. Unlike traditional methods that focus on either past research or future possibilities, this task incorporates both to generate informed projections, providing a more balanced perspective on scientific progress by ensuring that insights from the past inform future directions.

- **Coginitive-Inspired Framework**: Our approach *HorizonAI* is influenced by Dual-System Theory, suggesting that human cognition operates through both intuitive and analytical processes. In our framework, System 1 organizes past research into temporal knowledge graphs, while System 2 deliberately reasons to uncover patterns and anticipate future developments. This dual approach supports a more comprehensive, precise, and dynamic understanding of research trends.

- **Innovative Benchmark for Evaluation**: We propose a benchmark *ResBench* designed to evaluate *DTRA* based on historical coverage and predictive reliability. By incorporating datasets that span both historical and predictive dimensions, this benchmark provides the research community with a tool for systematically testing methods that summarize past research while forecasting future

developments. The integration of these two tasks enhances the performance of each, as insights from historical analysis inform predictions, leading to more accurate and contextually relevant forecasts.

## 2 DUAL-SYSTEM FRAMEWORK: *HorizonAI*

More formally, we define the *Dual Temporal Research Analysis*(*DTRA*) task as follows:
Given the *input topic* and *source paper* pair $(\mathcal{T}, \mathcal{P})$, the task is to narrate the *research history* $H$ of the topic and generate *possible research ideas* $F$.

To achieve this task, we propose *HorizonAI* (as illustrated in Figure 2), a framework inspired by Dual-System Theory. We retrieve related papers to represent the history $h_{s\sim t} = \{P_1, P_2, ..., P_n\}$ during the time interval of $s \sim t$ and structure it into a graph $\mathcal{G}$ (i.e. our *PaperTKG*), then using strategy $S$ to search the graph for timeline generation $\tau_{s\sim t} = S(\mathcal{G})$. The historical narrative $H$ is generated by LLMs using temporal reasoning based on the timeline. We sample possible future predictions $F$, $p(F|H) > threshold$ based on $H$.

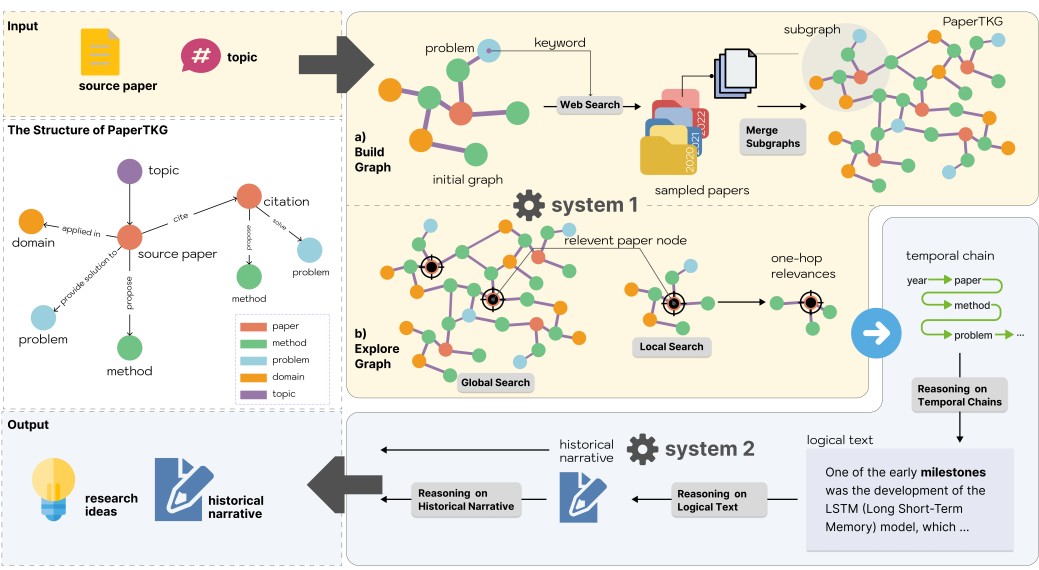

Figure 2: Diagram of *HorizonAI* framework. Given a topic and a source paper as input, *HorizonAI* goes through System 1 of a) structuring dynamically gathered historical information into *PaperTKG* and b) generating a timeline based on historical information and System 2 of reasoning on the timeline for historical narrative generation and future research trend prediction based on it.

### 2.1 *PaperTKG* CONSTRUCTION

To store historical information of research dynamically and structurally, we propose one specialized data structure—*PaperTKG*. It builds on the foundation of traditional TKGs by focusing specifically on academic papers. In *PaperTKG*, paper nodes are annotated with their timestamps and connected to entities such as methods, problems, domains, topics, and citations, as shown in the structure in Figure 2, storing all relevant and recent information to enhance reasoning in System 2 while reducing data processing costs.

The construction process of *PaperTKG* systematically progresses through three phases— building the initial graph, extending the graph by web search, and graph integration and refinement. The pseudo-code of our *PaperTKG* construction process can be found in Algorithm 1 and the prompts for it can be found in C.1.

**Paper2Graph** Converting papers to graphs (Paper2Graph) is a vital task in *PaperTKG* Construction. We use mainly the abstract and related work sections of a paper for that purpose. We derive the

---

**Algorithm 1** Temporal Knowledge Graph Construction

---

1: **Input:** Topic $T$, Source Paper $S$
2: **Output:** Temporal Knowledge Graph $G$
3: ***Phase 1:*** Build Initial Graph $G_0 = \text{Paper2Graph}(T, S)$
4: ***Phase 2:*** Extend Graph $G_0$
5: **for** each problem node $p$ in $G_0$ **do**
6:     **for** each year $y$ from $Y_{start}$ to $Y_{end}$ **do**
7:         search for related papers with keywords $\{T, p\}$
8:     **end for**
9: **end for**
10: **for** each new paper $P_i$, $i = 1$ to $N$ **do**
11:     $G_i = \text{Paper2Graph}(T, P_i)$
12: **end for**
13: ***Phase 3:*** Graph Integration
14: **for** each subgraph $G_i$, $i = 1$ to $N$ **do**
15:     **for** each problem node $p$ in $G_i$ **do**
16:         **for** each year $y$ from $Y_{start}$ to $Y_{end}$ **do**
17:             search for related papers with keywords $\{T, p\}$
18:         **end for**
19:     **end for**
20:     merge $G_i$ into $G_0$
21: **end for**
22: refine $G$: drop duplicates, discrete entities, complete missing entities
23: **return** $G$

---

problem, application domain, and proposed method from the abstract while related works section provides insights into connections between existing methods, problems, and domains, annotated by the authors (Inevitable subjective bias is addressed by bulk sampling of papers - See Appendix A.3). Algorithm 2 details the pipeline.

---

**Algorithm 2** Paper2Graph: Entity and Relation Extraction

---

1: **Input:** topic $T$, paper $P$
2: **Output:** subgraph $\mathcal{G}$ with core concepts from $P$ and its citations
3: ***Phase 0:*** *Citation Matching*
4: Match citations in related work to references
5: Create paper nodes and complete metadata via web search
6: ***Phase 1:*** *Local Extraction*
7: **for** each citation $c$ **do**
8:     Extract method, problem, and domain related to $c$ from context
9:     Establish entity relations
10: **end for**
11: ***Phase 2:*** *Overall Connection*
12: Infer relations between all entities
13: **return** $\mathcal{G}$

---

**Graph Augmentation** We extend graphs built by Paper2Graph by incorporating subgraphs from papers cited and papers retrieved through a targeted web search. To create a concentrated historical dataset, we adopt a problem-centric sampling strategy, using problem nodes as search keywords rather than querying databases directly. Initially, problem nodes guide the first sampling round, yielding a fixed number of papers per year (also called uniform sampling, see Appendix A.2 for further explaination). Each sampled paper is converted to a subgraph using Paper2Graph, with their problem nodes driving the second and final sampling round. For a source paper with $k_0$ problem nodes and $n_0$ citations, the first round adds $n_0 + L \cdot k_0 \cdot t$ subgraphs, sampling $t$ papers annually over $L$ years. Ultimately, we gather $k_0 + \sum_{i=1}^{n_0 + L \cdot k_0 \cdot t} k_i$ problems and $1 + n_0 + \sum_{i=1}^{n_0 + L \cdot k_0 \cdot t} n_i$ papers, ideally without duplication. To manage costs, we cap the total number of sampled papers.

## 2.2 Query Generation and Temporal Reasoning

**Query Generation and Graph Exploration** Inspired by the local-to-global search strategy for summarization utilized by the GraphRAG (Edge et al., 2024), we design a global-to-local search strategy to narrow down the search range step by step without missing related nodes or relations. We first use global search to locate paper nodes related to the query, then apply local search to get detailed relations and neighbors of the paper node. **Global Search:** The query for global search is generated from the following three aspects: the application domain, the target problem to solve, and the method. We traverse all the paper nodes and select the ones related to our query by similarity, then we check the timestamps of these nodes to ensure that representative works from each year are included. **Local Search:** In the local search phase, we collect the one-hop relevances of the target paper nodes (i.e. the details of the paper) and structure them into a chain, finally, we get the timeline for the topic.

**Temporal Reasoning** Large Language Models (LLMs) are utilized to perform temporal reasoning (Yuan et al., 2024) using Chain-of-Thought (CoT) prompting (Wei et al., 2022). They are prompted to identify key research milestones, explain the methods and solutions, highlight connections between works, and emphasize the progression over time, step by step, to generate a coherent narrative of the research history. The prompt for converting the timeline to logical text can be found in Appendix C.2.

## 2.3 Result Generation

The output of our *HorizonAI* consists of two components: a historical narrative and a future prediction, with the latter being generated based on the former.

**Historical Narrative** We narrate the history from both local and holistic perspectives using temporal reasoning through CoT prompts (listed in Appendix C.2). For the local perspective, we structure the narrative by using the subtitles from the related work sections of the target papers as an outline, producing content resembling related work discussions. For the holistic perspective, we create outlines based on section titles from selected surveys to represent the overall development of the topic. Each section is then expanded with content following the outline, resulting in a survey-like narrative.

**Future prediction** Existing approaches often emphasize the novelty of generated research ideas, overlooking that feasibility is a more critical factor than pure innovation. To ensure the ideas are grounded in practicality, we reference the local historical narrative and prompt LLMs to outline detailed roadmaps for realizing each idea. For each subdomain, multiple potential ideas are sampled, ensuring a balance between originality and implementability.

## 3 Proposed Benchmark: *ResBench*

### 3.1 Data Construction

Table 1: Topics Used in Data Collection.

| Index | Topic |
|-------|-------|
| 1 | In-context Learning |
| 2 | LLMs for Recommendation |
| 3 | LLMs-based Agents |
| 4 | Instruction Tuning for LLMs |
| 5 | LLMs for Information Retrieval |
| 6 | Safety in LLMs |
| 7 | Large Multi-Modal Language Models |
| 8 | LLMs for Software Engineering |
| 9 | LLM-Generated Texts Detection |

Our dataset comprises papers from the arXiv[1] repository, specifically focusing on LLMs. The dataset has 20 different topics, but considering the difficulty of manual verification, this article mainly evaluates 9 different topics (Table 1), covering the principles, techniques, and diverse applications of LLMs. For each topic, the dataset includes a source paper, a survey and at least 10 target papers. The source paper, an input for the task, serves as a starting point for collecting related literature to complete the historical information, while the surveys and target papers are used to evaluate the task outputs. In the surveys, every subsection's content and title are included, alongside the corresponding references and notable research contributions. More details of the data composition are shown in Appendix B.

---

[1] https://arxiv.org/

## 3.2 EVALUATION

The LLM evaluation consists of three main areas: historical completeness, predictive reliability ~~,and text readability~~ .

### 3.2.1 GRAPH COMPLETENESS AND SEARCH EFFICIENCY

The completeness of graphs is evaluated by comparing the overlap between citations in target surveys and paper nodes in TKGs. Similarly, the search efficiency of graphs is assessed by measuring the overlap between paper nodes in retrieved during search and those in the surveys. The following sets involved in our evaluation are defined (their relationships are illustrated in Figure 3):

- $R$: References in the surveys.
- $G$: Paper nodes in the constructed graphs.
- $S$: Paper nodes retrieved during graph search.
- $H$: Key historical works in the surveys.

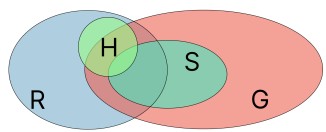

We define the following metrics to quantify the degree of overlap between these sets:

Figure 3: Venn Diagram. Note that S is a subset of G and H is a subset of R.

$$O_R = \frac{|G \cap R|}{|R|}, \quad O_H = \frac{|G \cap H|}{|H|}, \quad SE_R = \frac{|S \cap R|}{|G \cap R|}, \quad SE_H = \frac{|S \cap H|}{|G \cap H|}$$

Where $O$ stands for *Overlap*, evaluating graph completeness upon construction and $SE$ stands for *Search Efficiency*, evaluating graph search. More specifically:

- $O_R$: Average proportion of target paper citations present in the generated graph. Used to evaluate the graph completeness.
- $O_H$: Average proportion of key historical works present in the searched nodes. Used to evaluate the graph completeness, with a greater weight compared to $O_R$.
- $SE_R$: The ratio between searched historical works and all the historical work referenced in the surveys and constructed in the graphs. Used to evaluate the search efficiency.
- $SE_H$: The ratio between searched key historical works and all the key historical works referenced in the surveys and constructed in the graphs. Used to evaluate the search efficiency, with a greater weight compared to $SE_R$.

### 3.2.2 PREDICTIVE RELIABILITY

Predictive reliability is evaluated through four perspectives: semantic similarity $S_1$, innovation and feasibility $S_2$, temporal consistency $S_3$, and contextual consistency $S_4$. All values are rated by LLM based on prompt instructions (detailed in Appendix C.3.1) on a scale of 1 to 5. The final rating is a weighted sum of these values:

$$Final\_Score = w_1 \cdot S_1 + w_2 \cdot S_2 + w_3 \cdot S_3 + w_4 \cdot S_4$$

Where $w_1$, $w_2$, $w_3$, $w_4$ are weights for Semantic Similarity $S_1$, Innovation and Feasibility $S_2$, Temporal Consistency $S_3$, and Contextual Consistency $S_4$.

The explanation for the ranges of the final score is defined as:

- Final_Score $\in$ [1,2): The generated future directions show poor relevance to the target paper, with significant deficiencies in semantics, innovation, feasibility, or temporal consistency.
- Final_Score $\in$ [2,3): The generated future directions are somewhat relevant to the target paper but have several notable shortcomings.
- Final_Score $\in$ [3,4): The generated future directions are generally well-aligned with the target paper across multiple dimensions, though some improvements are still needed.
- Final_Score $\in$ [4,5): The generated future directions are highly relevant and excel in innovation, feasibility, temporal logic, and contextual consistency.

## 4 EXPERIMENTS

We use the GPT-4o (OpenAI, 2024a) for all the LLMs-involved processes (e.g. graph construction and reasoning) in our framework. We run experiments on the evaluation dataset on two subtasks for both our *HorizonAI* and baselines.

### 4.1 SUMMARIZING HISTORY - SURVEY COMPARISON

The performance of our history summarization subtask is assessed on the overlap degree of generated content and the target survey, whose result is illustrated in Table 7 where we also present the performance of graph completeness and search efficiency as a reference. As shown in Table 2, we compare the performance of our *HorizonAI* and AutoSurvey (Wang et al., 2024) in the history summarization subtask from three perspectives, namely total citation, key citation, and keyword. We conclude the performance of our framework as follows:

**Comprehensive historical representation**  Under the conditions of limited information and the presence of bias in the writing of the target survey, the paper nodes in our *PaperTKG* have an average 39.35% overlap ratio with the citations in the human-written surveys and an even higher score of 46.35% regarding key citation overlap, demonstrating that our method of structuring history into *PaperTKG* to thoroughly and logically arrange scientific history is effective.

**Efficient Search Strategy**  The average proportion of our searched works shared with the target survey among the total paper nodes of the graph reaches 69.92%, while on key references it is as high as 71.06%. The significantly high ratios of success indicate that using our global-to-local search strategy to fetch related paper nodes in the graph is powerful.

**Complete and Reliable History Summarization**  A small proportion of searched past works are lost after the reasoning phase. The final generated content has an average of 24.25% citations in common with the target survey, compared to 5.26% of AutoSurvey. It reveals that our *HorizonAI* has a better ability to trace and summarize influential past works.

Table 2: Comparison of our *HorizonAI* and AutoSurvey (Wang et al., 2024) in history summarization subtask evaluated on nine topics. We use the overlap ratio of two aspects—total citation and key citation—to evaluate the performance of this task.

| Evaluation Object | Citation Overlap(%) | | Key Citation Overlap(%) | |
|---|---|---|---|---|
| | *HorizonAI* (ours) | AutoSurvey | *HorizonAI* (ours) | AutoSurvey |
| Topic 1 | **42.86** | 5.44 | **53.01** | 12.12 |
| Topic 2 | **27.32** | 6.19 | **38.93** | 6.45 |
| Topic 3 | **2.27** | 2.27 | **37.87** | 10.81 |
| Topic 4 | **24.24** | 6.06 | **47.98** | 0.00 |
| Topic 5 | **27.93** | 4.14 | **46.19** | 6.98 |
| Topic 6 | **5.00** | 4.00 | **10.00** | 6.06 |
| Topic 7 | **35.57** | 4.35 | **50.00** | 12.50 |
| Topic 8 | **19.75** | 8.02 | **24.24** | 5.48 |
| Topic 9 | **33.33** | 6.86 | **25.40** | 8.33 |
| Average | **24.25** | 5.26 | **37.07** | 7.64 |

### 4.2 PREDICTING FUTURE - RELATED WORKS COMPARISON

We use the subtitles of related work of the target paper as a guideline to generate the possible research idea, then we compare this generated idea with the actual one proposed by this paper in the abstract. The performance of the future prediction is evaluated on the comprehensive score of the content, covering content quality, relevance, innovation, and so on. Due to the existing works aimed at idea generation mainly focusing on novelty, they will naturally filter out previous works. In response to this situation, we use LLM and horizonAI without temporal logic reasoning as our baseline to evaluate how much the performance of HorizonAI will drop without adopting a workflow inspired by dual system theory. The result of this subtask, as is illustrated in Table 3, proved that with adequate historical narrative and temporal logic reasoning, LLMs can produce more reliable research ideas than the ones without.

Table 3: Comparison between our *HorizonAI*, LLMs and *HorizonAI* without temporal logic reasoning in future prediction. The final score calculation method is shown in Section 3. The full score is 5.

| Evaluation Object | Baseline | Without Temporal Logic Reasoning | *HorizonAI* (ours) |
|---|---|---|---|
| Topic 1 | 3.77 | 2.30 | **3.91** |
| Topic 2 | 3.42 | 1.10 | **3.98** |
| Topic 3 | **3.88** | 1.25 | 3.85 |
| Topic 4 | 3.68 | 2.20 | **3.78** |
| Topic 5 | 3.50 | 2.05 | **3.76** |
| Topic 6 | 3.69 | 1.15 | **4.01** |
| Topic 7 | 3.44 | 2.00 | **3.87** |
| Topic 8 | 3.84 | 2.28 | **3.88** |
| Topic 9 | 3.23 | 1.90 | **3.91** |
| Average | 3.60 | 2.14 | **3.88** |

Table 4: Ablation for input quality. Problem type 1 stands for the interdisciplinary topics, and topics related to it are: A - Bias and Fairness in LLMs, B - LLMs in Medicine, C - Domain Specialization of LLMs, D - Challenges of LLMs in Education. Problem type 2 stands for the topic and source paper misalignment case, and the topic related to it is: E - Explainability for LLMs

| Problem Type | Topic | Source Paper | Number of Paper Nodes | History Completeness (%) | Citation Overlap (%) |
|---|---|---|---|---|---|
| | A | Gupta et al. (2023) | 44 | 2.99 | 1.45 |
| 1 | B | Singhal et al. (2023) | 996 | 6.19 | 2.44 |
| | C | Li et al. (2022) | 131 | 16.67 | 5.56 |
| | D | Leinonen et al. (2023) | 36 | 0.00 | 0.00 |
| 2 | E | Gao et al. (2023) | 620 | 2.68 | 1.52 |

## 4.3 ABLATION STUDY

**Effect of Input** We designed two possible problem types regarding the inputs to determine their influence on our method (as illustrated in Table 4). The first case involves interdisciplinary topics that require more relevant historical information compared to topics within the AI field. Additionally, obtaining related data from arXiv is relatively more challenging. Topics related to medicine, education, and society are selected for it. The results show that our method with a broad cross-domain topic as input suffers from graph augmentation failure, leading to an unwanted history completion performance. The second case involves a mismatch between the topic and source paper. In this case, a source paper with less relevance to the topic is given as input. This leads to the graph used for representing history expanding in the wrong direction, which explains the bad performance. In conclusion, our method is sensible to the inputs (i.e. the topic and the source paper), either a vague topic or mismatched inputs will lead to unwanted results.

**Effect of Graph Augmentation Strategy** We test the performance of history completeness under four different graph argumentation methods to determine the effect of web retrieval strategy on our framework. The complexity of collecting historical work increases sequentially from Method 1 to Method 4, with Method 4 being the one used in our framework. As shown in Figure 4, the performance variation trends of different search strategies across topics are consistent, and the more comprehensive the search method, the higher the citation overlap of the enhanced graph. Among them, Method 4 achieves the best performance across all topics. This result indicates that the graph argumentation method has a significant impact

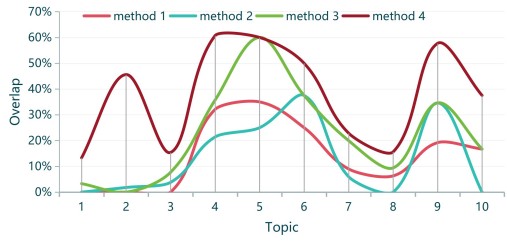

Figure 4: Diagram of search strategy performance. Method 1 is human efforts, Method 2 is greedy-search from the original problem in source paper, Method 3 uses similarity ranking to search from the original problem, while Method 4 (ours) updates on method 3 by searching on all central problems.

on completing historical data. More data does not necessarily mean better historical reproduction; rather, retrieving data from multiple dimensions is more beneficial for historical completion.

# 5 RELATED WORKS

## 5.1 RETRIEVAL ON KNOWLEDGE GRAPHS (KGS)

Recent search strategies using Knowledge Graphs (KGs) improve retrieval by leveraging structured relationships in Large Language Models (LLMs) to enhance inference and interpretability (Pan et al., 2024; Yang et al., 2024). Structuring LLM interactions with KGs refines retrieval performance, enabling effective responses to complex queries (Sun et al., 2023; Jiang et al., 2023; 2024). Unlike traditional RAG methods that rely on text embeddings, KGs serve as indices to enhance precision by navigating relevant subgraphs. Approaches like KAPING (Baek et al., 2023), G-Retriever (He et al., 2024), and Graph-ToolFormer (Zhang, 2023) enhance retrieval by using graph metrics to refine search results, while SURGE (Kang et al., 2023) and FABULA (Ranade & Joshi, 2023) leverage KGs for narrative generation grounded in factual subgraphs. Systems like ITRG (Feng et al., 2023) and IR-CoT (Trivedi et al., 2023) facilitate multi-hop question answering by tracing interconnected knowledge nodes, while Selfmem (Cheng et al., 2023) employs KGs for generation-augmented retrieval. GraphRAG (Edge et al., 2024) advances these approaches by introducing a unique local-to-global search strategy with a self-generated graph index, which inspired us for our global-to-local retrieval approach.

## 5.2 TEMPORAL KNOWLEDGE GRAPHS (TKGS)

Temporal Knowledge Graphs (TKGs) extend traditional Knowledge Graphs(KGs) by incorporating temporal information, enabling the representation of dynamically changing facts (Ji et al., 2021). Temporal Knowledge Graph Completion (TKGC) is a key task in TKGs, focusing on filling in missing information and predicting future relationships (Wang et al., 2023; Xu et al., 2023; Zhang et al., 2023; Xiong et al., 2024b). Additionally, TGKs specialize in applications like event tracking and historical data analysis, providing a more nuanced framework for mapping extensive academic literature. Specialized models such as Know-Evolve (Trivedi et al., 2017) and TA-TransE (García-Durán et al., 2018) have advanced temporal reasoning, while Xiong et al. (2024a) introduced a two-step framework for language-based temporal reasoning that translates narratives into TKGs. Despite these innovations, many existing models remain general and do not specifically address the organizational needs of academic information. Our proposed *PaperTKG* thus serves as a tailored TKG structure designed explicitly for managing scholarly papers, enabling the tracking of paper evolution, citation networks, and topic trends over time, thereby fulfilling the demand for a specialized TKG in academia.

## 5.3 LLMS IN SCIENTIFIC DEVELOPMENT

Large Language Models (LLMs) are recognized for their transformative potential in scientific research, owing to their ability to process and analyze vast datasets beyond human capacity. Recent studies, such as those by Baek et al. (2024), Yang et al. (2023), and Qi et al. (2023), focus on Literature-based Discovery (LBD) (Swanson, 1986), using LLMs to mine academic publications for correlations and generate research insights. Wang et al. (2024) explores the possibility of LLMs automatically generating survey papers, while other works (Elsevier, 2024; Agarwal et al., 2024) emphasize automated retrieval and summarization of existing literature, often neglecting the prediction of future research trends. In a pioneering effort, Li & Flanigan (2024) formalizes future language modeling, aiming to predict future textual data based on temporal histories. Additionally, several studies (Si et al., 2024; Baek et al., 2024; Zheng et al., 2024) develop LLM-based agents for research idea generation, a critical step in the early stages of scientific inquiry. AI Scientist (Lu et al., 2024) represents the first comprehensive system for fully automated scientific discovery using LLMs, generating novel research ideas independent of prior work, though it requires multiple iterations to yield viable outcomes. In contrast, we introduce *HorizonAI*, a dual-system approach that integrates both the summarization of past research and the prediction of future directions, offering superior performance in both tasks compared to existing models.

# 6 CONCLUSION

In this paper, we introduced *Dual Temporal Research Analysis* (*DTRA*), a novel task that integrates the summarization of historical research with the prediction of future trends. Our framework, *HorizonAI*, draws inspiration from Dual-System Theory to organize past research using *PaperTKG* (a temporal knowledge graph for papers) and employs LLMs for in-depth reasoning to generate both historical narratives and future projections.

Through extensive evaluation on the *ResBench* benchmark, we demonstrated that bridging the tasks of historical analysis and future forecasting enhances the performance of both. Our results showed significant improvements in summarizing past works and generating accurate predictions compared to existing methods.

The integration of historical insights with predictive reasoning offers a balanced perspective on scientific progress, showing the potential of *HorizonAI* as a robust tool for supporting research across multiple domains. Future work will focus on expanding the dataset beyond AI-related topics and enhancing search capabilities to incorporate a wider range of academic databases.

# 7 LIMITATION AND FUTURE WORKS

## 7.1 LIMITATIONS

1. The dataset currently focuses on AI-related topics with surveys available in 2024, but it can be extended to a broader range of domains. This design was chosen to facilitate more precise evaluation and easier expert feedback, but future work should include diverse research fields to enhance generalizability.

2. Currently, the search and graph construction processes are time-consuming due to the reliance on third-party web APIs that often struggle with bulk access. This issue can be addressed by using specialized API keys or developing our own databases. Nevertheless, the current system still offers higher efficiency compared to manual research, and the constructed graphs can be reused for further analysis.

3. The current assessment of future idea generation relies solely on LLMs in content analysis; however, while the accuracy of utilizing our algorithm is guaranteed, the inclusion of expert reviewers will provide additional insight into the feasibility and reliability of the ideas generated.

## 7.2 FUTURE WORKS

In addressing previous limitations, we encourage extending the dataset beyond AI-related topics to include a broader range of research fields. This expansion would allow for a more comprehensive evaluation of the framework's generalizability across different domains. Additionally, we plan to enhance our data collection by incorporating papers from other sources beyond Arxiv, such as peer-reviewed journals and other preprint servers, using advanced tools for PDF information extraction. Furthermore, integrating expert reviews into the evaluation process will provide more reliable insights into the feasibility and practical relevance of the generated future ideas, moving beyond sole reliance on LLM evaluations.

On the other hand, we will continue to explore ways to enhance research efficiency in the era of LLMs and AI. This area holds significant potential, and beyond generalization and future direction prediction, we aim to enable AI to contribute to the actual realization of future research topics. This will involve collaboration with researchers in experiment design and result analysis, integrating AI more deeply into the research process.

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

# A FURTHER CLARIFICATION ON STRATEGY

## A.1 HOW DO WE REPRESENT RESEARCH HISTORY?

There are various ways to define the history of research, but academic papers remain one of the most common forms of scholarly communication. A typical method for scientists to understand the evolution of a field is by reviewing related papers. Thus, we define the history of a field or research question as the collection of papers associated with it:

$$H = \{P_1, P_2, \ldots, P_n\} \tag{1}$$

where $P_i$ denotes the $i$-th paper in the collection, and $n$ is the total number of relevant papers.

To expand this collection more effectively, we extract papers from the related works sections:

$$H' = \bigcup_{i=1}^{n} RW(P_i) \tag{2}$$

where $H'$ represents the extended history, and $RW(P_i)$ is the set of cited works in $P_i$.

The progression of knowledge in a field can be represented as a trajectory, with papers ordered temporally:

$$T = \{P_{\sigma(1)}, P_{\sigma(2)}, \ldots, P_{\sigma(n)} \mid t_{\sigma(1)} \leq t_{\sigma(2)} \leq \cdots \leq t_{\sigma(n)}\} \tag{3}$$

where $\sigma$ is a permutation of indices ensuring the papers are arranged chronologically.

To represent this trajectory more efficiently, we use temporal knowledge graphs, detailed further in Section 3 of the Methods.

## A.2 HOW DO WE SAMPLE FROM HISTORICAL PAPERS?

Sampling is essential to ensure the accuracy and diversity of research findings, mitigating bias. Table 5 compares several sampling strategies: *Uniform Sampling*, *Proportional Sampling*, *Citation-based Sampling*, *Random Sampling*, and *Stratified Sampling*. We select *Uniform Sampling* due to its ability to maintain an even distribution across years, minimizing variance and offering ease of implementation.

Table 5: Overview of Sampling Methods and Their Variance. The other methods have $> 0$ variance influenced by factors such as publication volume, citation counts, and strata representation. $P(p_i)$ is the probability of selecting papers $p_i$, $N_j$ is the number of papers in a time period $y_j$, $N$ is the total number of papers, $C_i$ is the citation count of paper $p_i$, $M$ is the total sample size, and $m_j$ is the sample size for each period or stratum.

| Sampling Method | Mathematical Definition | Pros | Cons | Variance in $m_j$ |
|---|---|---|---|---|
| **Uniform Sampling** | $P(p_i) = \frac{1}{|P_{y_j}|},$ $\forall i \in y_j$ | Balanced representation across time periods | May exclude influential papers from prolific years | 0 |
| Proportional Sampling | $P(p_i) = \frac{N_j}{N},$ $m_j = P(p_i) \cdot M$ | Reflects natural publication volume | Over-represents years with high publication counts | $> 0$ |
| Citation-based Sampling | $P(p_i) = \frac{C_i}{\sum_{P \in P_{y_j}} C_P}$ | Focuses on highly influential papers | Skews toward older papers; Ignores recent work | $> 0$ |
| Random Sampling | $P(p_i) = \frac{1}{N}$ | Simple, unbiased by time or citation | May miss important trends; Over-represents recent years | $> 0$ |
| Stratified Sampling | $P(p_i) = \frac{N_j}{N},$ $m_j = P(p_i) \cdot M$ | Ensures representation across strata | Complex to implement; Over-represent dominant strata | $> 0$ |

## A.3 Addressing Subjective Bias in Publications

Subjective bias is inherent in individual papers, as each presents knowledge from a particular viewpoint. Consequently, integrating biased papers into a study introduces this subjectivity. However, by aggregating enough diverse papers, we can mitigate individual biases and approach a more objective historical representation:

$$B(H) = \sum_{i=1}^{n} B(P_i) \tag{4}$$

To reduce bias, we aim to incorporate a sufficiently large set of papers. The bias in an expanded collection $H'$ of papers can be approximated by:

$$B(H') \approx \frac{1}{|H'|} \sum_{i=1}^{|H'|} B(P_i) \tag{5}$$

As the size of $H'$ increases, the overall bias approaches a more balanced representation of the field.

## B   DATA COLLECTION

The data collection process focused on identifying relevant papers across three categories: **source papers**, **target papers**, and **surveys**. These papers were selected based on their influence, citation count, and relevance to the topic, ensuring a comprehensive overview of the field.

### B.1   SOURCE PAPERS

Source papers refer to highly cited and influential papers published before **2023**. These papers were carefully chosen based on their significant contributions to the field and their role in shaping foundational knowledge. Each source paper was analyzed for its related work sections, which included:

- The titles of cited references.
- Summaries of key points from these references.

We selected these papers using Semantic Scholar's influential sorting feature (Kinney et al., 2023), ensuring the source papers were ranked by citation count. Data was extracted from HTML and PDF formats, with arXiv[2] and ar5iv[3] providing the HTML versions for most papers. All data underwent manual verification to ensure accuracy.

### B.2   TARGET PAPERS

Target papers refer to newer, high-quality papers and surveys from **2024**, chosen for their cutting-edge insights. These papers were similarly ranked by **citation count** and were selected to reflect the most current trends and advancements in the field. In target papers, we also focused on their related work sections, capturing:

- The titles of cited references.
- Key points and summaries relevant to the topic.

Target papers helped bridge the gap between historical research and the latest developments, providing a forward-looking perspective.

### B.3   SURVEYS

Surveys were treated as a separate category, as they provide an overview of the field and summarize key developments. For surveys, we included every subsection's content and title, alongside corresponding references. In addition, we focused on identifying **notable research contributions**, defined as:

- Articles or works that were frequently cited.
- Papers described with extensive detail by the survey authors, often corresponding to key subheadings.

These typically represent key historical works and important research results such as the development of technologies like Transformers.

Surveys were instrumental in identifying key historical works (*key_history*) that had a lasting impact on the field. These works were defined by their influence and the significant number of references made to them within the surveys.

More details of the data composition are shown in Table 6.

---

[2]https://arxiv.org/
[3]https://ar5iv.labs.arxiv.org/

Table 6: Details of the data information.

| Entity Type | Content | Example |
|---|---|---|
| topic | | "in-context learning" |
| year_start | | "2021" |
| year_end | | "2024" |
| source_paper
↓


reference(full) | name, arxiv_id,
isAPA, abstract,
reference,
related_work | {"name": "Chain-of-Thought...", "arxiv_id": "2201.11903",
"isAPA": true, "abstract": "We explore how generating...",
"reference": [ reference1, reference2,... ], "related work":
"7Related Work...", "date": "2022"}
"Subhro Roy and Dan Roth. 2015. Solving general arithmetic
word problems. EMNLP" |
| target_list
↓

target_paper
↓

subtitle

reference | name, arxiv_id,
subtitles, reference,
related_work | [target_paper1, target_paper2,...]

{ "name": "Long-context LLMs...", "arxiv_id":
"2404.02060", "subtitles": [subtitle1, subtitle2,...],
"reference":[reference1, reference2,...] , "related_work":
"2Related Work..." }
"Reinforcement Learning via Supervised Learning (RvS)"

"Eva: Exploring the limits of masked visual representation
learning at scale" |
| survey
↓

reference
subtitle(full)
↓

key_history | name, arxiv_id,
subtitles,
all_references


name, key_history,
refer-
ences_in_this_section
reference_title,
key_word | {"name": "In-context Learning...", "arxiv_id": "2401.11624",
"subtitles": [subtitle1, subtitle1,...], "all_references":
[reference1, reference2,...]}
"Eva: Exploring the limits of masked visual representation
learning at scale"
{"name": "Few-shot...", "key_history": [key_history1,
key_history2,...] , "references_in_this_section": [reference1,
reference2,...]}
{"reference_title": "Attention is all you need", "key_word":
"Transformer Models"} |
| topic_history
↓

reference | name, arxiv_id,
reference | { "name": "Long-context LLMs...", "arxiv_id":
"2404.02060", "reference":[reference1, reference2,...] }
"Eva: Exploring the limits of masked visual representation
learning at scale" |

# C PROMPTS

## C.1 PROMPTS FOR GRAPH CONSTRUCTION

**Extract from Abstract**

```
EXTRACT_THEME = '''Please extract the key issue addressed, the proposed
    method, and the application domain from the abstract of the paper
    titled *{title}* and present the information in the following JSON
    format.
{{
    "problem": {{
        "name": the key issue it addressed,
        "description": a more detailed description of this key issue
    }}
    "method": {{
        "name": the method it proposed,
        "description": a more detailed description of this method
    }}
    "domain": {{
        "name": the application domain,
        "description": a more detailed description of this domain
    }}
}}]
If any of the information is not available, please fill the corresponding
     value with 'null'. Note that the descriptions should be extracted
    from context, DO NOT simply use your prior knowledge to complete them
    .
Absract Content: {abstract}'''
```

**Extract from related works**

```
LEVEL1 = '''Please extract the method and problem entities related to the
     citation '{citations}' from the excerpt of the paper titled '{title
    }', and identify the relations between these entities and the
    citation. Please respond with the following JSON format.
        [
            {{
                "entity name": The name of the entity that has relation
                    with the citation '{citations}'. DO NOT extract human
                     names as entities,
                "entity type': The type of the entity, selected from '
                    method', 'problem', and 'domain',
                "description": Description of the entity extracted from
                    the context, null if not enough information,
                "relation": The relationship between the citation and the
                     entity extracted from the context can be expressed
                    using phrases such as 'applied in', 'proposed by',
                    and others. Ensure that the relation is explicitly
                    mentioned in the text and avoid inferring any
                    relations based on prior knowledge. Do not use vague
                    description like 'related to'
            }},
            ...
        ]
    '''
LEVEL2 = '''Find out the relationships between these entities in the
    content. DO NOT add relations including entities that do not exist in
     the list. Please respond with the following format.
    [{{
        "entity1": The name of the entity1,
        "relation": The relationship between entity1 and entity2. Ensure
            that the relation is explicitly mentioned in the text and
            avoid inferring any relations based on prior knowledge. Do
            not use vague description like 'related to',
```

```
1026        "entity2": The name of the entity2
1027    }},
1028    ...]
1029    Entities: {entities}
1030    Content: {content}'''
```

## C.2 PROMPTS FOR REASONING

**Generate Related Works**

```
generate_relatedwork_prompt = f"""
Let's generate a high-quality "Related Work" section for a research paper
    by following a structured reasoning approach. We will use the
    following steps to ensure clarity and depth.
**Step 1: Analyze the topic and the narrative's progression.**
The topic is '{topic}', and the subtitle is '{subtitle}'. Here is the
    time-based progression of research developments:\n\n{cot_narrative}\n
    \n
Analyze the key themes, shifts, and milestones in the narrative to
    extract the most relevant and impactful works that shaped the field
    over time.
**Step 2: Identify key studies.**
Based on the analysis, identify the most influential and representative
    studies that have contributed to the advancement of this field.
    Select works that either introduced foundational concepts, solved
    critical challenges, or advanced the field in significant ways.
**Step 3: Structure the Related Work section.**
Organize the selected studies in a way that emphasizes their contribution
     to the progression of the field. The section should naturally flow
    either chronologically or thematically, ensuring a balance between
    foundational works and recent innovations. You may highlight any gaps
     or ongoing debates in the literature to contextualize how these
    works relate to your research.
Now, based on this reasoning, generate the "Related Work" section. Make
    sure it is flexible but retains a coherent narrative that aligns with
     academic standards. Incorporate key research areas and their
    evolution in the field, using a mix of foundational works and recent
    studies. The section should demonstrate a clear understanding of how
    these works interrelate and how they contribute to the current
    research landscape.
Please provide the response in "Related Work" section only, structured as
     follows:
"""
example="""
**Related Work**
The field of {main_topic} has evolved significantly over the past few
    decades, particularly in areas such as {key_areas_1}, {key_areas_2},
    and {key_areas_3}. Early works such as {Author1 et al., Year} laid
    the groundwork for {specific concept or technique}, introducing key
    methods that have since been built upon by later studies.
For instance, *{Key Area 1}* has been a major focus, starting with
    foundational research by {Author2 et al., Year}, who proposed {a
    major contribution}. Building on this, subsequent studies like {
    Author3 et al., Year} have refined these approaches, introducing
    innovations such as {specific advancement} that have made a
    substantial impact in the field.
In contrast, *{Key Area 2}* represents a more recent development, with
    groundbreaking contributions by {Author4 et al., Year}, who explored
    {a novel approach or finding}. This has opened new avenues for
    research, particularly in {specific application or challenge}, as
    evidenced by {Author5 et al., Year}, whose work has further expanded
    on these ideas.
Additionally, the intersection of *{Key Area 3}* with {related field} has
     also gained attention in recent years. Notably, {Author6 et al.,
```

```
    Year} demonstrated {key contribution}, which has been instrumental in
      advancing the understanding of {specific problem or question}.
While much progress has been made, there remain open questions,
    especially in {specific area of ongoing research}, where recent
    studies by {Author7 et al., Year} indicate that further exploration
    is needed to fully realize the potential of {key technique or
    approach}.
By reviewing these works, we gain a comprehensive understanding of how
    the field has evolved and where it is headed, providing essential
    context for the contributions of our own research.
"""
generate_relatedwork_prompt+=example
```

**Generate Future Idea**

```
Now, let's  think step by step to generate predictions for future
    research directions based on the provided time-based narrative and
    subtitle.

Step 1: **Analyze the time-based narrative for future trends**. The
    narrative '{cot_narrative}' shows how the research has evolved over
    time. Carefully examine this narrative to extract clues about current
     trends, technological bottlenecks, and potential gaps in research
    that could drive future developments.

Step 2: **Consider challenges and opportunities**. Based on the trends
    and patterns identified in the narrative, think about the challenges
    the field currently faces and the opportunities for future
    innovations. What are the key bottlenecks, and what cutting-edge
    technologies could overcome them?

Step 3: **Predict future research directions**. Based on the analysis,
    predict possible future directions for the topic '{topic}' and
    subtitle '{subtitle}'. These directions should be logically derived
    from the observed research trends and potential advancements in
    technology.

Step 4: **Structure the future directions and technical roadmap**.
    Organize the predicted future research directions into a well-
    structured roadmap, clearly outlining the steps researchers might
    take to advance in this area. Present the future directions in JSON
    format.
Please provide the response in JSON format only, structured as follows:
Example output format:
```json
{{
  "Future_Directions": {{
    "1. Title of Future Direction": {{
      "Description": "Detailed description of the future research
          direction, derived from trends in the narrative.",
      "Technical_Roadmap": [
        "First step in technical roadmap, based on observed trends.",
        "Next steps, reflecting future possibilities derived from the
            narrative."
      ]
    }},
    "2. Another Future Direction": {{
      "Description": "Another future research direction logically derived
           from current research challenges and gaps.",
      "Technical_Roadmap": [
        "First step, addressing challenges seen in the narrative.",
        "Subsequent steps reflecting the roadmap towards technological
            advancements."
      ]
    }}
```

```
  }}
}}
```
```

### C.3   PROMPTS FOR EVALUATION

#### C.3.1   PROMPT FOR FUTURE RELIABILITY EVALUATION

$S_1$: **Semantic Similarity**

```
The following is the abstract of a target paper:\n\n{target_abstract}\n\n
    Below is a set of future research directions predicted by another
        paper:\n\n{future_text}\n\n
    Your task is to carefully compare the abstract and the predicted
        future directions step by step using chain of thought reasoning.

    Step 1: Analyze the future directions and extract key research themes
         or topics.
    Step 2: Compare each theme with the abstract to identify any matching
         concepts.
    Step 3: Assign a score based on the extent of matching as follows:

    - **Score 0**: No matches at all; none of the key themes or topics
        from the future directions are present in the abstract.
    - **Score 1**: Very few matches; only 1 out of 5 key themes match
        with the abstract.
    - **Score 2**: Some matches; 2 out of 5 key themes match with the
        abstract.
    - **Score 3**: Moderate matches; 3 out of 5 key themes match with the
         abstract.
    - **Score 4**: Mostly matches; 4 out of 5 key themes match with the
        abstract.
    - **Score 5**: All matches; all 5 key themes from the future
        directions are present in the abstract.

    Please **only return the final score as a single number**.
```

$S_2$: **Innovation and Feasibility**

```
The following is a future research direction proposed by a research paper
    :

    "{future_direction}"

    Please evaluate the following aspects step by step:
    Step 1: Analyze the future direction to determine if it proposes a
        novel or innovative idea compared to current research in the
        field, specifically related to the topic: {topic} and subtitle: {
        subtitle}. Does it introduce new concepts, techniques, or
        approaches that are not commonly explored?
    Step 2: Assess the technical feasibility of this future direction.
        Can it be realistically implemented with current technology, or
        does it require significant breakthroughs?
    Step 3: Rate the future direction on a scale from 1 to 5:
    - 1: No innovation and technically infeasible.
    - 2: Slight innovation but mostly infeasible.
    - 3: Moderately innovative and feasible with technical challenges.
    - 4: Innovative and feasible with current technology, with minor
        challenges.
    - 5: Highly innovative and technically feasible without significant
        challenges.

    Please **only return the final score as a single number**.
```

$S_3$: **Temporal Consistency**

```
The following is a future research direction proposed by a research paper
    :

    "{future_direction}"

    Please evaluate the following step by step:
    Step 1: Analyze the current state of research and technological
        progress in the relevant field of {topic}, and related to the
        subtitle: {subtitle}. Identify the key milestones and major
        developments up to now.
    Step 2: Determine if this future direction logically builds upon
        recent developments, or if it requires an unrealistic leap
        forward in technology.
    Step 3: Assess whether the future direction aligns with the current
        pace of technological development. If it seems unrealistic for
        the near future, explain why.
    Step 4: Rate the temporal consistency of the future direction on a
        scale from 1 to 5:
    - 1: Does not fit the timeline at all.
    - 2: Slightly inconsistent with the timeline.
    - 3: Moderately consistent with the timeline, with some gaps.
    - 4: Largely consistent with minor inconsistencies.
    - 5: Fully consistent with the timeline and logically follows current
         research progress.

    Please **only return the final score as a single number**.
```

### $S_4$: Contextual Consistency

```
The following is a future research direction proposed by a research paper
    :

    "{future_direction}"

    The target paper's abstract is as follows:

    "{target_abstract}"

    Please evaluate the following aspects step by step:
    Step 1: Identify the key research challenges or limitations discussed
         in the target paper's abstract, which relates to the topic: {
        topic} and subtitle: {subtitle}. What are the primary issues the
        target paper seeks to address?
    Step 2: Determine if the proposed future direction addresses any of
        these challenges or builds upon the research presented in the
        target paper.
    Step 3: Assess whether the proposed future direction logically
        follows the research context or is disconnected from the
        challenges identified in the target paper.
    Step 4: Rate the contextual consistency of the future direction on a
        scale from 1 to 5:
    - 1: Completely disconnected from the research context.
    - 2: Slightly relevant, but mostly misaligned with the research
        context.
    - 3: Moderately related to the research context, but missing key
        connections to the identified challenges.
    - 4: Largely consistent, addressing most of the research challenges
        with minor gaps.
    - 5: Fully aligned with the research context, addressing key
        challenges comprehensively.

    Please **only return the final score as a single number**.
```

### C.3.2 Weight settings for Future Reliability Evaluation

Weight Distribution: In our future prediction evaluation task, we use the following weights:

( $w_1 = 0.4$ ) for historical completeness and prediction reliability (this includes manual evaluation), ( $w_2, w_3, w_4 = 0.2$) each for the LLM-based assessments of prediction reliability, text generation quality, and other factors. We assign the highest weight ( $w_1 = 0.4$ ) to historical completeness because this portion involves manual evaluation, which we believe is more accurate and reliable, particularly for complex tasks such as evaluating the completeness of the graph and the reliability of future predictions. Manual evaluation offers higher credibility compared to the more automated LLM assessments.

On the other hand, the LLM-based evaluations are given lower weights because they rely on automated models, and their results can be more dynamic and fluctuate depending on the context, input, and other factors. While LLM assessments are valuable for scalability, we acknowledge that their results are not as stable or trustworthy as human assessments in these particular tasks.

# D    RESULT SPECIFICS

## D.1    HISTORICAL SUMMARIZATION EVALUATION RESULTS

The specific values for our evaluation metrics of the nine topics can be viewed below:

Table 7: The performance of history completeness and history summarization. We utilize the overlap degree of paper nodes in the graph and citations in the survey to evaluate the completeness of history. The search phase is a preliminary process of history summarization, we show the search efficiency on both overall and key citations to indicate the reasoning performance. The history summarization performance is assessed on the overlap degree of generated content and the target survey. See the definition of the metrics in Section 3.

| Evaluation Object | Graph Completeness(%) | | Search Efficiency(%) | | Generated Content |
|---|---|---|---|---|---|
| | $O_R$ | $O_H$ | $SE_R$ | $SE_H$ | Overlap Degree(%) |
| Topic 1 | 50.00 | 65.74 | 75.71 | 74.51 | 42.86 |
| Topic 2 | 30.00 | 46.00 | 63.00 | 79.00 | 27.32 |
| Topic 3 | 70.00 | 73.00 | 43.00 | 36.00 | 2.27 |
| Topic 4 | 57.00 | 38.00 | 81.00 | 68.00 | 24.24 |
| Topic 5 | 34.00 | 59.00 | 77.00 | 73.00 | 27.93 |
| Topic 6 | 21.00 | 35.00 | 17.00 | 19.00 | 5.00 |
| Topic 7 | 36.00 | 50.00 | 97.00 | 100.00 | 35.57 |
| Topic 8 | 25.00 | 25.00 | 79.00 | 90.00 | 19.75 |
| Topic 9 | 31.13 | 25.40 | 96.54 | 100.00 | 33.33 |
| Average | 39.35 | 46.35 | 69.92 | 71.06 | 24.25 |

## D.2    CASE STUDY ON FUTURE PREDICTION: TOPIC 2 - LLMS FOR RECOMMENDATION

**Future Directions Generated by Our *HorizonAI***
The ratings are 4 for $S_1$, 4 for $S_2$, 4.17 for $S_3$, 3.75 for $S_4$, with an average of 3.98.

1. **Enhancing Generative Recommendations with LLMs** *Description*: Building on the paradigm-shifting survey on generative recommendations, focus on overcoming efficiency challenges and improving the ethical integration of LLMs in recommender systems. *Technical Roadmap*: 1. Develop optimized algorithms to reduce computational overhead in LLM-based generative recommendations. 2. Explore pruning and quantization techniques to make LLMs more efficient for real-time applications. 3. Conduct extensive studies on the ethical implications of LLM recommendations, formulating guidelines for ethical usage.

2. **Integrating User Mobility Patterns in Recommendations** *Description*: Leverage LLMs to better understand and predict user mobility patterns, enhancing recommendation relevance in location-based services. *Technical Roadmap*: 1. Create datasets that capture diverse mobility patterns and integrate them with user activity data. 2. Develop LLM-based models that can learn and predict user mobility patterns to provide context-aware POI recommendations. 3. Evaluate the effectiveness of mobility-pattern integrated recommendations in real-world scenarios and refine models accordingly.

3. **Cross-Domain Knowledge Transfer in Recommender Systems** *Description*: Address the limitations of single-domain pre-training by exploring techniques for cross-domain knowledge transfer using LLMs. *Technical Roadmap*: 1. Investigate methodologies for transferring knowledge between different domains to enhance recommendation capabilities. 2. Develop frameworks to assess the quality and applicability of transferred knowledge across domains. 3. Implement and test cross-domain transfer models in various recommendation contexts to determine best practices.

4. **Enhancing Behavioral Simulation for Recommendation Development** *Description*: Refine the agent-based simulation methods introduced by RecAgent to create even more sophisticated user behavior models. *Technical Roadmap*: 1. Expand user and recommender modules to capture a broader range of user behaviors and interac-

tions. 2. Integrate reinforcement learning techniques to continuously improve simulated user-adaptive responses. 3. Validate simulated environments against real-world data to ensure the fidelity and applicability of simulations.

5. **Ethical and Fair Recommendation Systems** *Description*: Further research into developing frameworks for ensuring fairness, transparency, and ethical considerations in LLM-powered recommender systems. *Technical Roadmap*: 1. Develop metrics and benchmarks to evaluate fairness and transparency in recommendation systems. 2. Create auditing tools to detect and mitigate biases in LLM-generated recommendations. 3. Formulate policies and best practices to uphold ethical standards in deploying LLM-based recommenders.

6. **Real-Time Adaptability in LLM-Based Recommender Systems** *Description*: Focus on making LLM-based recommender systems more adaptive in real-time scenarios to provide timely and context-sensitive recommendations. *Technical Roadmap*: 1. Develop techniques for rapid model updates and fine-tuning based on real-time user feedback and environmental changes. 2. Implement scalable infrastructure to support real-time adaptability without compromising performance. 3. Test real-time adaptable systems in diverse recommendation applications and refine models for robustness.

**Future Directions Generated by GPT-4o**

The ratings are 3.6 for $S_1$, 3.3 for $S_2$, 3.8 for $S_3$, 2.98 for $S_4$, with an average of 3.42.

1. **Integrating LLMs with Geographic Information Systems (GIS) for Next-POI Recommendation** *Description*: Develop sophisticated models that combine the power of LLMs with GIS to provide highly accurate next-POI recommendations, leveraging spatial and temporal data. *Technical Roadmap*: 1. Develop methods to preprocess and integrate GIS data with LLM inputs, ensuring data coherence and compatibility. 2. Enhance LLM capabilities to understand geographical nuances and contexts by fine-tuning on spatial datasets. 3. Create hybrid models that combine LLM predictions with GIS analytics, implementing validation mechanisms specific to location-based predictions. 4. Deploy and iteratively improve the recommendation system using real-world data and feedback loops.

2. **Personalization in LLM-Based Recommender Systems** *Description*: Address personalization challenges by developing LLMs capable of deeply understanding individual user preferences and historical behavior for tailor-made recommendations. *Technical Roadmap*: 1. Aggregate extensive user interaction data and develop pre-training tasks that reflect typical user behaviors. 2. Investigate transfer learning methods to adapt general LLMs to specific user profile data, enhancing personalization accuracy. 3. Integrate reinforcement learning techniques to continuously adapt and improve recommendations based on real-time user feedback. 4. Evaluate and implement privacy-preserving mechanisms to ensure user data protection while personalizing recommendations.

3. **Predicting User Mobility Patterns Using LLMs** *Description*: Leverage LLMs to predict complex user mobility patterns by integrating various data sources like social media, transportation logs, and historical movements. *Technical Roadmap*: 1. Identify and unify diverse data sources (e.g., social media, GPS logs) to create comprehensive mobility datasets. 2. Design pre-training tasks focused on mobility pattern recognition to enhance LLM understanding of movement data. 3. Implement sequence-based LLM architectures to capture temporal dynamics and predict future movements accurately. 4. Develop evaluation frameworks to assess the predictive performance and refine models based on predictive accuracy and reliability metrics.

4. **Predicting User Mobility Patterns Using LLMs** *Description*: Leverage LLMs to predict complex user mobility patterns by integrating various data sources like social media, transportation logs, and historical movements. *Technical Roadmap*: 1. Identify and unify diverse data sources (e.g., social media, GPS logs) to create comprehensive mobility datasets. 2. Design pre-training tasks focused on mobility pattern recognition to enhance LLM understanding of movement data. 3. Implement sequence-based

LLM architectures to capture temporal dynamics and predict future movements accurately. 4. Develop evaluation frameworks to assess the predictive performance and refine models based on predictive accuracy and reliability metrics.

5. **Cross-Domain Implications of LLM-Based Recommendation Systems** *Description*: Explore the applicability and implications of LLM-based recommender systems across various domains (e.g., retail, entertainment, health) to uncover new opportunities and challenges. *Technical Roadmap*: 1. Conduct domain-specific studies to understand the unique requirements and constraints of LLM applications in different sectors. 2. Develop adaptable LLM architectures that can efficiently switch contexts and deliver domain-specific recommendations. 3. Implement cross-domain transfer learning techniques to enhance LLM generalizability while preserving domain-specific nuances. 4. Continuously monitor and document the performance, ethical considerations, and user satisfaction across these diverse applications.

