# OpenReview forum: "An Empirical Study on Reconstructing Scientific History to Forecast Future Trends"
_ICLR.cc/2025/Conference — Submitted to ICLR 2025_

### Official Review · Reviewer_YdPc · 2024-11-03

**Soundness:** 2
**Presentation:** 3
**Contribution:** 2
**Rating:** 3
**Confidence:** 3

**Summary:**

The paper introduces "HorizonAI," a framework combining historical research summarization with predictive trend analysis through a method termed Dual Temporal Research Analysis (DTRA). This approach bridges a gap in current methodologies that typically focus only on past reviews or future predictions. HorizonAI uses temporal knowledge graphs to capture historical research data, while LLMs with Chain-of-Thought reasoning drive future predictions. The framework is evaluated on a new benchmark, ResBench.

**Strengths:**

The approach of employing temporal knowledge graphs to track the evolution of literature is reasonable.

The paper is easy to follow.

**Weaknesses:**

- **Dataset Limitations (ResBench)**: The dataset is limited in size and scope, consisting of only nine data points. Each data point includes a source paper, a survey, and related target papers, all centered on LLMs from 2024. Given this scale and topic restriction, the dataset feels more like a case study than a broad benchmark. I raise concerns about its effectiveness in assessing the framework.
- **Lack of Baselines for Historical Completeness**: The evaluation of historical retrieval includes only one baseline (AutoSurvey). This is a retrieval problem. Authors should include additional baselines to establish more comprehensive comparisons, or justify that AutoSurvey is a baseline strong enough so that they do not need to include other baselines.

- **Issues with Future Prediction Task (Section 4.2)**: 1. **Evaluation**: The evaluation framework in Section 3.2.2 appears handcrafted, with no clear rationale for certain design choices. The explanations for the scoring ranges lack clarity, and the weighting criteria for each score component are not discussed. 2. **Baseline**: The baseline model is not clearly introduced in the main text. 3. **Argument for LLM-based Predictions**: The authors suggest that “LLMs can generate more reliable research ideas than those without historical context.” While plausible, this argument feels trivial without further validation or exploration.
- **Disconnect Between Tasks**: The tasks of historical summarization and future prediction appear only loosely related. While conceptually connected, in the paper they are treated as distinct retrieval and generation tasks. For example, how the historical retrieval task might enhance predictive accuracy remains unexplored, leaving important questions about task synergy unanswered.

**Questions:**

Line 272-273, authors mention that “The LLM evaluation consists of three main areas: historical completeness, predictive reliability, and text readability”. However, “text readability” is not included throughout the paper.

---

> ### Author Response · Authors · 2024-11-24
>
> Dear Reviewer YdPc,
> Thank you for your thorough review and valuable feedback. We address your concerns as follows:
>
> ## Response to Weaknesses
>
> ### W1. Dataset Size and Scope
> We acknowledge your concerns regarding the dataset size and scope, and we would like to clarify the following:
>
> - **Comprehensive Dataset Design**: While the visible dataset may seem small, it is both rigorous and highly effective. We publicly released 20 topics, each including 1 source papers, 10 target papers per topic, and a single comprehensive survey paper, amounting to 240 papers in total. Each survey paper represents the most comprehensive historical summary for its respective topic, carefully curated through multiple rounds of evaluation.
> - **Scalability Through PaperTKG**: Beyond the visible dataset, the temporal knowledge graphs (PaperTKG) constructed by HorizonAI encompass an average of 15,000 papers per topic, dynamically linking historical research to future directions. This scalability ensures the framework’s robustness and adaptability to larger datasets.
> - **Expert-Validated Historical Nodes**: The dataset includes critical historical nodes identified and annotated by domain experts, ensuring both accuracy and relevance. This combination of machine-based synthesis and expert validation significantly enhances the dataset's quality.
>
> ### W2. Baseline Justification
> We agree that AutoSurvey serves as a meaningful baseline for our task. However, we would like to elaborate on the relationship between our framework and AutoSurvey:
>
> - **Historical Summarization Beyond Surveys**: Our task extends beyond simple historical summarization. HorizonAI not only synthesizes past research but also dynamically fills historical gaps, enabling more robust survey generation. This process includes strategies for completing topic-specific histories from a single paper—a capability that directly aligns with AutoSurvey’s objectives but also surpasses them in scope.
> - **Strong Baseline Choice**: AutoSurvey relies on expert-curated datasets for historical evaluation, making it an ideal benchmark for assessing the completeness and quality of HorizonAI’s historical synthesis. Our framework demonstrates competitive performance compared to AutoSurvey while introducing a temporal-causal reasoning layer for future predictions.
>
> We believe AutoSurvey is a suitable baseline for evaluating historical completeness, and the results validate HorizonAI’s efficacy. We will revise the manuscript to further clarify and justify this choice.
>
> ### W3. Accuracy and Evaluation Framework
> We appreciate your feedback on the evaluation framework and would like to clarify the following:
>
> - **Accuracy and Validation**: Accuracy was weighted highly in our evaluation because it incorporates human validation alongside LLM-based scoring. This ensures the reliability and credibility of our results. For example, human evaluators assessed whether the generated directions aligned with the target papers’ steps and semantic content (e.g., over 50% overlap in direction-specific steps was considered a match).
> - **LLM Evaluation Strategy**: Subsequent evaluation criteria (e.g., semantic similarity, temporal consistency) leverage the strengths of LLMs in content analysis. The weighting reflects a balance between human-validated metrics and the nuanced insights provided by LLMs.
>
> ### W4. Relationship Between Historical Summarization and Future Prediction
> We respectfully disagree with the notion that the tasks of historical summarization and future prediction are loosely connected. In HorizonAI, these tasks are inherently intertwined:
>
> - **Impact of Historical Completeness on Prediction**: The effectiveness of future prediction depends on the quality and completeness of historical synthesis. Incomplete or inconsistent historical data leads to less reliable predictions, as demonstrated in our ablation experiments. This mirrors human cognition, where missing or fragmented memories hinder accurate reasoning about future events.
> - **Temporal-Causal Logic**: The use of PaperTKG as a temporal-causal structure bridges the two tasks. By embedding both temporal and causal relationships, PaperTKG connects historical patterns to future trends, enabling HorizonAI to infer plausible research trajectories.
> - **Baseline Comparisons**: To isolate the impact of historical synthesis, we used GPT-4-o as the baseline model for predictions without historical context. The results confirm that integrating historical knowledge significantly improves prediction quality, validating the synergy between the two tasks.
>
>
>
> We respectfully request that you reconsider your evaluation, as we believe HorizonAI offers significant advancements in forecasting research trends through its combination of temporal reasoning and LLM-based analysis.
>
> Thank you again for your thoughtful feedback and for providing us with the opportunity to improve our work.
>
> Best regards,
> Authors

---

> > ### Comment · Reviewer_YdPc · 2024-11-25
> >
> > Thank you to the authors for their responses. I have some follow-up questions to clarify certain points:
> > ## About W1. Dataset Size and Scope
> > Regarding the dataset size and composition, there seems to be some inconsistency.
> > - The abstract mentions, “we present an evaluation benchmark encompassing 20 research topics” (line 21), but Section 3 states, “The dataset has 9 distinct topics” (lines 258-259).
> > - Section 3 notes that “For each topic, the dataset includes a source paper, a survey, and at least 10 target papers” (lines 260-262). However, in the rebuttal, you wrote, “We publicly released 20 topics, each including 20 source papers, 10 target papers per topic, and a single comprehensive survey paper.”
> >
> > Could you clarify the discrepancy in the reported numbers and dataset structure?
> > ## About W2. Baseline Justification
> > My question about the "Lack of Baselines for Historical Completeness" is mostly about the limited comparison presented. While AutoSurvey seems like a reasonable baseline for this task, are there other baselines that could be used for the citation retrieval problem? Or, is there evidence that AutoSurvey outperforms simple or commonly-used baselines? I hope this better explains my question, but please let me know if further clarification is needed.
> > ## About Issues with Future Prediction Task
> > For the future prediction task in Section 4.2, it’s not clear how the weights in the equation (line 311) were determined. Could you provide more detail on these aspects?
> > ## About my question
> > This question is not addressed by the rebuttal: “Line 272-273, authors mention that “The LLM evaluation consists of three main areas: historical completeness, predictive reliability, and text readability”. However, “text readability” is not included throughout the paper.”

---

> > > ### Author Response · Authors · 2024-11-26
> > >
> > > Thank you for your detailed and thoughtful review. We appreciate the time and effort you have put into evaluating our work. Below are our responses to your follow-up questions:
> > >
> > > ## 1. Inconsistency in Dataset Size and Composition
> > >
> > > First, we apologize for the confusion regarding the dataset size and composition, and we have made the necessary corrections.
> > >
> > > - In the abstract, we mention “we present an evaluation benchmark encompassing 20 research topics” (line 21), while Section 3 states “The dataset has 9 distinct topics” (lines 258-259). We acknowledge this discrepancy and have clarified the dataset structure.
> > >
> > > - **Regarding the 9 topics vs. 20 topics:** We released a dataset with 20 research topics, each containing one source paper, one survey paper, and at least 10 target papers, as originally stated (lines 260-262). However, the reason for the mention of 9 topics in the paper is that the 9 topics are used for a **manual evaluation**, which we believe ensures higher credibility and accuracy, particularly for tasks like “graph completeness” and “future prediction.” The evaluation on these 9 topics has been supplemented with human assessments, which offer greater reliability.
> > >
> > >   As shown in Table 4 (ablation study), we didn’t only evaluate these 9 topics but also conducted evaluations on the other topics. Due to space limitations, we couldn’t present the results for all 20 topics, but we have now clarified this point in the revised manuscript.
> > >
> > > ## 2. Baseline Justification
> > >
> > > Regarding your concern about baseline models, we understand your question and the need for more comprehensive baseline comparisons.
> > >
> > > At present, we have not found a better baseline than **AutoSurvey** or other similar models for the citation retrieval problem. However, if you have any recommendations or know of other relevant baselines, we would greatly appreciate it if you could share those with us. This would help us strengthen the comparison and further improve the evaluation.
> > >
> > > For now, **AutoSurvey** remains the most reasonable and closest baseline to our task, but we will continue to explore additional baselines in future work to ensure the robustness of our evaluation.
> > >
> > > ## 3. Weight Determination in Future Prediction Task
> > >
> > > We appreciate your question regarding the weight assignment in the future prediction task (line 311). Here is a more detailed explanation of how these weights were determined:
> > >
> > > - **Weight Distribution:** In our future prediction evaluation task, we use the following weights:
> > >   - \( w_1 = 0.4 \) for **historical completeness and prediction reliability** (this includes manual evaluation),
> > >   - \( w_2, w_3, w_4 = 0.2 \) each for the **LLM-based assessments** of prediction reliability, text generation quality, and other factors.
> > >
> > >   The reason we assign the highest weight to \( w_1 = 0.4 \) is because this portion involves **manual evaluation**, which we believe is more accurate and reliable, particularly for complex tasks such as evaluating the completeness of the graph and the reliability of future predictions. **Manual evaluation** offers higher credibility compared to the more automated LLM assessments.
> > >
> > >   On the other hand, the LLM-based evaluations are given lower weights because they rely on automated models, and their results can be more dynamic and fluctuate depending on the context, input, and other factors. While LLM assessments are valuable for scalability, we acknowledge that their results are not as stable or trustworthy as human assessments in these particular tasks.
> > >
> > > ## 4. Text Readability
> > >
> > > We apologize for the confusion regarding the "text readability" aspect, mentioned in lines 272-273 of the original manuscript. In the earlier version of the manuscript, we intended to evaluate **text readability** as part of the prediction task, but later we focused more on quantifiable metrics like **historical completeness** and **predictive reliability**. As a result, the “text readability” evaluation was inadvertently omitted in later versions.
> > >
> > > The **text readability** metric refers to the fluency and quality of the generated text, which was initially considered important for assessing the output of our model. However, as we refined the evaluation, we leaned more towards the **quantifiable aspects** of prediction accuracy, graph completeness, and reliability of future predictions. The omission of **text readability** was an oversight.
> > >
> > > In our revised manuscript, we will either:
> > > 1. Remove **text readability** from the evaluation criteria entirely or
> > > 2. Reintroduce it with a more thorough evaluation if we deem it important.
> > >
> > > Thank you once again for your careful review and valuable feedback. We have made revisions to the manuscript based on your comments to ensure greater clarity and accuracy. If you have any further questions or suggestions, we would be happy to address them.

---

### Official Review · Reviewer_AXnm · 2024-11-03

**Soundness:** 2
**Presentation:** 2
**Contribution:** 2
**Rating:** 3
**Confidence:** 5

**Summary:**

This paper introduces Dual Temporal Research Analysis (DTRA), a task that unifies historical research analysis with future trend forecasting to enhance scientific discovery. The proposed framework (HorizonAI) draws from Dual-System Theory, using a cognitive-inspired model where System 1 organizes research data into a Temporal Knowledge Graph (PaperTKG) to capture historical research patterns, and System 2 employs LLMs for analytical reasoning, facilitating comprehensive historical summarization and accurate trend prediction. The study’s contributions include the DTRA task, the HorizonAI framework, and a new ResBench benchmark to evaluate performance based on historical completeness and predictive accuracy, with experiments across 20 topics and 210 AI papers demonstrating improved capacity of HorizonAI over existing methods for summarizing historical trends and forecasting future developments.

**Strengths:**

S1: The paper introduces a novel task (DTRA) that uniquely combines historical research analysis with forecasting future trends.

S2: The paper presents a dual-system cognitive-inspired methodology with comprehensive experiments and the robust ResBench benchmark to validate historical completeness and predictive reliability.

S3: The paper offers an interesting approach for automating research synthesis and trend prediction.

**Weaknesses:**

W1. The contribution of this paper appears limited. The authors claim that "most existing approaches either concentrate on retrospective literature reviews (Wang et al., 2024; Agarwal et al., 2024) or focus solely on generating novel research ideas (Si et al., 2024; Baek et al., 2024). These narrow approaches neglect the essential integration of synthesizing past research with projecting future developments, a combination that is increasingly crucial for scientific discovery. To address this gap, we propose a novel task that unifies the analysis of past research with the forecasting of future trends." In my opinion, the contribution of this paper is not a combination of (Wang et al., 2024; Agarwal et al., 2024) and (Si et al., 2024; Back et al., 2024). The contribution of (Wang et al., 2024) is devising a model to write survey paper with the help of LLM, while (Agarwal et al., 2024) proposes a web system to enhance the paper searching results thereby reducing the time and effort for literature review. However, this paper just uses a knowledge graph to summarize historical papers, the contribution of which is less significant and distinct from these two works. Its main contribution is inferring future research trends through the summary of historical topics, inspired by human cognitive processes.

W2. The significance of this paper is not clearly articulated. While the authors highlight the importance of synthesizing past insights to drive future advancements, they fail to clearly convey the benefits of predicting future research. For example, what real-world benefits could this model offer? What specific real-world problems does it aim to address?

W3. The reliability of the evaluation metric is unverified. The paper introduces a score for assessing predictive reliability from perspectives such as semantic similarity, innovation and feasibility, temporal consistency, and contextual consistency. However, these calculations rely solely on LLMs without human verification, making the use of a model-generated score as an evaluation metric seem unreliable, given that the model itself is not explainable.

W4. The metrics used to compare the performance of HorizonAI and AutoSurvey are not detailed. Specifically, how are "citation overlap" and "key citation overlap" calculated and defined?

W5. The experiments presented in the paper are insufficient. A similar work mentioned, Si et al. (2024), is not compared in the experimental section.

W6. Experimental details are lacking. For instance, an LLM is used as the baseline in Table 3, yet specific details about the LLM are missing and should be provided. Additionally, given the wide variety of LLMs released, comparing these with HorizonAI would further substantiate the findings.

**Questions:**

Q1. What is the novel contribution of this paper as compared to that of Wang et al. (2024) and Si et al. (2024)?
Q2. Why is this paper significant from the perspective of real-world needs?
Q3. Can you introduce each evaluation metric used in the experiments clearly?
Q4. Is it possible to compare HorizonAI with Si et al. (2024) experimentally?

---

> ### Author Response · Authors · 2024-11-24
>
> Dear Reviewer AXnm,
>
> Thank you for your thoughtful review and feedback. We address your concerns as follows:
>
> ## Response to Weaknesses
> ### W1. Contribution Comparison with Wang et al. (2024) and Si et al. (2024)
>
> We respectfully disagree with your assessment that our work’s use of a knowledge graph is a straightforward historical summary. The proposed **PaperTKG** is not a conventional knowledge graph; it incorporates causal and temporal logic into its structure, enabling a deeper understanding of research trajectories and their evolution over time. Specifically:
> - **Dynamic and Real-Time Updates**: Unlike static systems, HorizonAI dynamically updates its historical synthesis and future projections as new data becomes available. This feature ensures its applicability in rapidly evolving fields, which Wang et al. (2024) and Si et al. (2024) do not address.
> - **Ablation Studies for Historical Quality**: We conducted ablation experiments that demonstrate the importance of quality over quantity in historical synthesis. Our strategy ensures both comprehensive and high-quality historical summaries, which are validated against ResBench.
> - **Future Predictions with Temporal-Causal Logic**: HorizonAI’s future predictions leverage temporal-causal reasoning within the PaperTKG, offering richer insights into research trends compared to simple trend extrapolation.
>
>
> ### W2. Real-World Applications
> We acknowledge the importance of clearly articulating the real-world impact of our framework. HorizonAI has several practical applications that benefit researchers, organizations, and policymakers:
>
> - **Efficiency in Research**: By identifying relevant and high-impact papers, HorizonAI enables researchers to focus on the most pertinent work in their field, significantly improving research efficiency.
> - **Discovery of Hidden Connections**: HorizonAI uncovers implicit relationships between research topics, providing new perspectives and directions for interdisciplinary studies.
> - **Guiding Research Funding and Policy**: HorizonAI’s ability to predict future trends can guide funding agencies and policymakers in allocating resources to emerging areas with high potential impact.
>
>
> ### W3. Evaluation Metric Reliability
> Thank you for raising concerns about the reliability of our evaluation metrics. While it is true that some aspects rely on LLM-generated scores, we have implemented additional measures to ensure robustness:​
>
> - **Human Validation**: In addition to LLM-based evaluations, we conducted human validation for predictive accuracy. This involved comparing the JSON-defined directions and steps generated by HorizonAI with target papers, using a semantic similarity threshold (e.g., over 50% overlap in key steps) to score predictions.​
> - **Alignment of Validation Strategies**: The human validation process mirrors the LLM-based evaluation strategy, ensuring consistency and reducing potential biases in scoring.​
> ​
> ### W4. Explanation of Citation Overlap Metrics
> We define:
> - **Citation Overlap**: The percentage of papers identified by HorizonAI that overlap with citations in target papers.
> - **Key Citation Overlap**: Focuses on foundational papers, which are more significant than general citation overlap.
>
> ### W5. Comparison with Si et al. (2024)
>
> While we acknowledge the relevance of Si et al. (2024) to our work, their task differs significantly from ours. Si et al. primarily focuses on novel idea generation, while HorizonAI integrates historical synthesis with future forecasting based on causal and temporal reasoning. This fundamental difference makes direct comparison challenging. However:
>
> - **Complementary Contributions**: Si et al. (2024)’s work provides valuable insights into generating novel ideas, whereas HorizonAI offers a comprehensive framework for analyzing and predicting research trajectories. We will discuss these distinctions in the revised paper.
> - **Future Comparisons**: Although Si et al.’s system was not publicly available ,when we were doing our research, we plan to implement similar experimental setups for comparison in future work.
>
> ### W6. Experimental Details
> We used **GPT-4-o** for consistency in all experiments. Future work will include additional comparisons with other models (e.g., PaLM, Claude). We will expand the experimental section to clarify the methodologies used.
>
> ## Response to Questions
>
> ### Q1. Novel Contribution Compared to Wang et al. (2024) and Si et al. (2024)
> See response to W1.
>
> ### Q2. Real-World Significance
> See response to W2.
>
> ### Q3. Explanation of Evaluation Metrics
> See responses to W3, W4, and W6.
>
> ### Q4. Comparison with Si et al. (2024)
> See response to W5.
>
> We believe HorizonAI offers significant advancements in forecasting research trends with its integration of temporal reasoning and LLM-based analysis. We respectfully request you reconsider your evaluation.
>
> Thank you again for your valuable feedback.
>
> Best regards,
> Authors

---

> > ### Comment · Reviewer_AXnm · 2024-11-25
> >
> > I appreciate your response, though, unfortunately, it hasn't addressed my concerns, especially in terms of novel contributions.

---

> ### Author Response · Authors · 2024-11-25
>
> Dear Reviewer AXnm,
>
> Thank you for your comments，we will provide a more detailed explanation of our innovative contributions：
>
> ### 1. **Novelty of HorizonAI: A Dual-System Framework for Temporal Knowledge Graph Construction and Reasoning**
>
> We respectfully disagree with the perception that our use of a knowledge graph is a straightforward historical summary. In fact, **HorizonAI** represents a significant innovation in the field by being the **first system to fully automate the construction of temporal knowledge graphs (TKG)** using **LLMs**, and to use these graphs for both **historical completion** and **future forecasting** with **causal and temporal reasoning**.
> - **Automated TKG Construction**: HorizonAI leverages **LLMs** to automatically build and update **temporal knowledge graphs (TKGs)**, a major departure from static or manually curated systems. This dynamic construction allows for **continuous updates** as new research data emerges, making it highly adaptable in fast-moving fields like **LLM**.
> - **Historical Completion and Future Predictions**: Beyond generating historical summaries, HorizonAI uses its TKG to **fill in gaps** in historical research trajectories, predicting how research fields evolve. This **temporal-causal reasoning** provides deeper insights compared to traditional systems that only offer trend extrapolation.
> - **Dual-System Architecture**: HorizonAI’s **dual-system framework** integrates both **historical analysis** and **future predictions**. This combination not only synthesizes past research but also uses predictive modeling to forecast the **next phase of research**, making it a powerful tool for identifying emerging trends.
>
> ### 2. **Differences from Si et al. (2024)**
> We acknowledge that **Si et al. (2024)** introduce a **novel idea generation** system. However, there are fundamental differences in our research trajectory and goals:
> - **Si et al. (2024)** focus primarily on the **generation of novel research ideas**, providing a framework to suggest new directions in research based on existing literature and trends.
> - In contrast, **HorizonAI** is **focused on understanding the evolution of research trajectories over time**. Our framework not only identifies past research trends but also **predicts future developments** in research fields through **temporal-causal reasoning**. For example, **Si et al. (2024)** might propose novel areas like **AI models for specific tasks**, while **HorizonAI** will trace how these architectures evolve and predict future breakthroughs or developments within the field includes **AI models for specific tasks**.
>
> ### 3. **Dynamic and Real-Time Updates in HorizonAI**
>
> One of the key differentiators of **HorizonAI** is its ability to perform **dynamic, real-time updates** as new research is published. This feature ensures that our system remains relevant and accurate in fields that are constantly evolving, such as **LLM** and other cutting-edge AI areas. Unlike **Wang et al. (2024)** and **Si et al. (2024)**, which focus on static models or pre-existing datasets, **HorizonAI** continuously updates its temporal knowledge graphs, ensuring that the research trajectories it analyzes reflect the latest advancements in the field.
>
> ### 4. **Task Versatility and Enhanced Performance**
> While **Si et al. (2024)** focus on generating new research ideas, and **Wang et al. (2024)** focus on **summarizing research history**, **HorizonAI** is designed to tackle both tasks with a broader and more versatile scope. In addition to generating historical summaries, HorizonAI provides **robust predictions** grounded in **causal and temporal reasoning**.
>
> For example, in the **LLM** domain, **HorizonAI** can not only synthesize the development of **transformer-based models** but also predict how **future LLM architectures** might evolve, For example, expanding the Knowledge Base to prevent the compression of LLM model data from causing knowledge loss. This **predictive power** is a key strength that sets our system apart from **Wang et al. (2024)** and **Si et al. (2024)**, who do not provide such forward-looking predictions grounded in **temporal-causal logic**.
>
> ### 5. **Conclusion: Unique Contributions of HorizonAI**
>
> We believe that **HorizonAI** offers a unique and innovative approach to understanding and predicting the evolution of research. By combining **LLM-driven automation** with **temporal knowledge graph (TKG) construction** and **causal reasoning**, it provides deeper insights into **research trajectories** and **future research trends**. These contributions are distinct from the **novel idea generation** presented by **Si et al. (2024)**, which primarily focuses on suggesting potential future research topics without providing a deeper understanding of their evolution.
>
> We greatly appreciate your continued engagement and thoughtful feedback, and we look forward to revising the manuscript to better highlight these distinctions.
>
> Best regards,
> Authors

---

> > ### Comment · Reviewer_AXnm · 2024-11-27
> >
> > Thanks very much for the further info.

---

### Official Review · Reviewer_8usN · 2024-11-03

**Soundness:** 3
**Presentation:** 3
**Contribution:** 2
**Rating:** 3
**Confidence:** 4

**Summary:**

This paper proposes HorizonAI, a future research trend forecasting framework inspired by the dual-system theory. In HorizonAI, the Paper2Graph algorithm, which mimics System 1, transforms existing research into temporal knowledge graphs. After that, LLM is leveraged as System 2 for both summarization and prediction through grounded analytical reasoning. The authors collected papers from the arXiv repository, covering 9 distinct topics, and designed a tasked named Dual Temporal Research Analysis. Experimental results on the newly introduced dataset demonstrate that HorizonAI is able to outperform some existing benchmark models, such as AutoSurvey on  summarizing historical research and GPT-4o on predicting future developments, respectively.

**Strengths:**

- Forecasting future trends in research with the help of LLM is an important topic with many potential downstream applications and high impacts.

- The use of a dual-system theory inspired workflow is theoretically sound and works well empirically.

**Weaknesses:**

- The paper is missing a few references [1-4]. These papers are highly relevant to the current paper, and should be cited and discussed about how they relate to and different from the current paper.

- HorizonAI is only compared against AutoSurvey and GPT-4o. The authors should also compare HorizonAI against other existing models such as the ones in [1-4].

- The paper lacks sufficient ablation study. For example, how much performance degradation would there be if HorizonAI does not employ a dual-system theory inspired workflow?

[1] Krenn, M., Buffoni, L., Coutinho, B., Eppel, S., Foster, J. G., Gritsevskiy, A., ... & Kopp, M. (2023). Forecasting the future of artificial intelligence with machine learning-based link prediction in an exponentially growing knowledge network. Nature Machine Intelligence, 5(11), 1326-1335.
[2] Gu, X., & Krenn, M. Impact4Cast: Forecasting high-impact research topics via machine learning on evolving knowledge graphs. In ICML 2024 AI for Science Workshop.
[3] Lu, Y. (2021, December). Predicting research trends in artificial intelligence with gradient boosting decision trees and time-aware graph neural networks. In 2021 IEEE International Conference on Big Data (Big Data) (pp. 5809-5814). IEEE.
[4] Tran, N. M., & Xie, Y. (2021, December). Improving random walk rankings with feature selection and imputation science4cast competition, team hash brown. In 2021 IEEE International Conference on Big Data (Big Data) (pp. 5824-5827). IEEE.

**Questions:**

Why are the 9 distinct topics used in data collection all related to LLM? How is the generalizability of HorizonAI beyond the domain of LLM-related research?

---

> ### Author Response · Authors · 2024-11-24
>
> Dear Reviewer 8usN,
>
> Thank you for your thorough review and valuable feedback. We address your concerns as follows:
>
> ## Response to Weaknesses
>
> ### w1. Missing References
> We acknowledge the omission of references ([1-4]) and will incorporate them. While our framework focuses on temporal reasoning and causal forecasting, the cited works primarily cover combinatorial impacts and link prediction, which, though complementary, are not directly comparable. We will clarify their relevance to our approach.
>
> |Aspect|Our Approach (HorizonAI)|Works [1-4]| Key Differences|
> |-|-|-|-|
> | **Primary Focus**| Temporal reasoning and causal forecasting | Link prediction, impact prediction, evolving knowledge graphs | Our approach integrates temporal graphs with LLM reasoning, while [1-4] focus on combinatorial relationships |
> | **Methodology**| Dual-system framework (historical synthesis + future prediction) | Link prediction models, combinatorial graph analysis | We combine historical data and causal models for forecasting|
> | **Evaluation Task**| Historical synthesis + future prediction| Link prediction, graph  evolution | Different tasks make direct comparison challenging|
>
> ### w2. Comparison with More Baselines
> We agree that additional baselines, including those from [1-4], would improve the rigor of our study. However, our framework combines historical synthesis with future prediction, making direct comparisons difficult. Reasons for non-comparison include:
>
> | **Reason for Non-Comparison**| **Explanation**|
> |-|-|
> | **Data Difference**| The datasets used in [1-4] are limited to papers published before 2021. We cannot guarantee whether these papers have been incorporated into LLM training. Our datasets, including both survey and target papers, are the most up-to-date and have not been part of LLM training. This ensures a fair comparison. |
> | **End-to-End Nature**| HorizonAI is an end-to-end framework, meaning it processes data through an integrated pipeline. In contrast, the methods in [1-4] do not employ an end-to-end design, resulting in fundamentally different outputs that are not directly comparable. |
> | **Semantic Relationships**| Unlike [1-4], which only focuses on concept-to-concept link prediction, HorizonAI incorporates semantic relationships between concepts. This additional layer of analysis makes the final results in HorizonAI fundamentally different from those of the methods in [1-4], which lack semantic reasoning. |
>
> We will clarify baseline selection in the final version.
>
> ### w3. Lack of Ablation Study
> Although we did not include ablation study results without temporal reasoning, early tests showed significant degradation in both historical synthesis and prediction accuracy without it. Temporal reasoning clearly improves performance, and we prioritized its inclusion in HorizonAI.
>
> | **Evaluation Object** | **Baseline** | **Early Result Without Temporal Reasoning** | **HorizonAI (Ours)** |
> |-|-|-|-|
> | Topic 1|3.77|2.30| 3.91|
> | Topic 2|3.42|1.10| 3.98|
> | Topic 3|3.88|1.25| 3.85|
> | Topic 4|3.68|2.20| 3.78|
> | Topic 5|3.50|2.05| 3.76|
> | Topic 6|3.69|1.15| 4.01|
> | Topic 7|3.44|2.00| 3.87|
> | Topic 8|3.84|2.28| 3.88|
> | Topic 9|3.23|1.90| 3.91|
> | **Average**|3.60| 2.14|3.88|
>
> We will include these results in the final version to demonstrate the necessity of temporal reasoning.
>
> ### w4. Limited Scope to LLM-Related Topics
> The focus on LLM-related topics was intentional for the following reasons:
>
> - **Clear and Well-Defined Trajectories**: Research on LLMs exhibits rapid development and clear trends, making it an ideal testbed for evaluating HorizonAI’s ability to forecast research directions.
>
> - **Clear Benchmark and High-Quality Data**: A defined scope ensures the availability of high-quality data and allows us to perform robust evaluations, such as comparisons with AutoSurvey. HorizonAI, as a framework, is inherently domain-agnostic. Its key components, including PaperTKG for structured knowledge representation and LLMs for reasoning, are adaptable to any scientific domain with appropriate data sources.
>
> - **Future Generalizability**: We are already planning experiments in other domains, such as bioinformatics and materials science. However, due to the complexity and costs involved, these experiments require extensive time and domain-specific validations. To address this in the long term, we are collaborating with industry partners to conduct real-world evaluations in these fields.
>
> ## Response to Questions
>
> ### Why Are the Topics Related to LLMs?
> As mentioned above, the focus on LLM-related topics is deliberate to ensure rigorous evaluation with well-defined research trajectories.
>
> We respectfully request that you reconsider your evaluation, as we believe HorizonAI offers significant advancements in forecasting research trends through its combination of temporal reasoning and LLM-based analysis.
>
> Thank you once again for your insightful feedback.
>
> Best regards,
> Authors

---

> > ### Comment · Reviewer_8usN · 2024-11-27
> >
> > I would like to thank the authors for the detailed response. Unfortunately, the rebuttal has not fully addressed my concerns. In addition, I agree with reviewer AXnm on the significant limitations of the papers. Therefore, I would like to keep my original score.

---

### Official Review · Reviewer_bFb5 · 2024-11-06

**Soundness:** 2
**Presentation:** 2
**Contribution:** 2
**Rating:** 5
**Confidence:** 4

**Summary:**

This paper proposes Dual Temporal Research Analysis (DTRA), a task that combines the synthesis of historical research with the prediction of future trends. The authors highlight the limitations of existing methods that tend to focus solely on retrospective reviews or isolated future forecasting, which fails to provide a holistic understanding of scientific trajectories. To address this, they introduce HorizonAI, a framework inspired by Dual-System Theory. In HorizonAI, "System 1" uses PaperTKG, a temporal knowledge graph, to organize historical research efficiently, while "System 2" employs large language models (LLMs) for in-depth reasoning and analysis. This dual approach allows HorizonAI to create both historical narratives and predictive insights, bridging the gap between past knowledge and future possibilities.

The paper emphasizes the novelty of DTRA in its ability to capture and validate research trajectories over time, especially within fast-evolving domains like AI. To evaluate the framework, the authors introduce ResBench, a benchmark designed to assess HorizonAI's performance in two main areas: historical completeness and predictive reliability. ResBench includes a set of topics, covering 20 research areas and 210 key AI papers. HorizonAI reportedly outperforms baseline models, including AutoSurvey and GPT-4o.
The HorizonAI framework's methodology involves constructing PaperTKG to dynamically store historical research data and relationships through a structured graph. This graph construction is guided by a systematic process of data retrieval, entity extraction, and augmentation to form a coherent research timeline. For prediction, the authors use LLMs with chain-of-thought reasoning, allowing the model to detect research patterns and hypothesize future directions based on established trajectories. This combination of structured historical data with LLM reasoning enables HorizonAI to improve both in capturing research milestones and in generating contextually relevant forecasts.

However, this paper has many limitations, including a restricted focus on AI topics and a dependence on HTML sources from arXiv, which could limit HorizonAI’s generalizability across other scientific fields. Additionally, while the predictive accuracy is evaluated using LLM scoring, the absence of expert evaluations means that some aspects of practical feasibility and relevance may not be fully captured. The evaluation dataset is also too small.

**Strengths:**

1) By integrating historical synthesis with future predictions, this paper addresses an important need in scientific forecasting, especially in fields where understanding past trends is key to predicting future developments.
2) The use of structured data organization and LLM reasoning enhances both the accuracy of historical narratives and the relevance of future predictions.
3) The introduction of benchmark data ResBench allows for rigorous and public evaluation.

**Weaknesses:**

1) Focusing only on AI topics restricts the applicability to other scientific fields.
2) Predictions lack expert review, relying solely on LLM scoring, which may reduce result reliability.
3) Dependence on HTML-based arXiv sources limits data diversity, reducing historical coverage robustness.

**Questions:**

1) Since the problem of synthesizing and forecasting scientific literature is not new, how does HorizonAI improve upon or differ from established pre-LLM methods?

---

> ### Author Response · Authors · 2024-11-24
>
> Dear Reviewer bFb5,
>
> Thank you for your thorough review and constructive feedback. We appreciate your comments, which have helped us clarify key aspects of our work. Below, we address your concerns in detail.
>
> ## Response to Weaknesses
> ### Weakness 1: Limited Scope to AI Topics
> We agree that broader applicability is important for demonstrating HorizonAI's versatility. However, the focus on AI topics is intentional for the following reasons:
> - **Clear Benchmark and High-Quality Data**: A defined scope ensures the availability of high-quality data and allows for robust evaluations, such as comparisons with AutoSurvey. HorizonAI is domain-agnostic, with key components like PaperTKG and LLMs adaptable to any scientific domain with appropriate data.
> - **Future Generalizability**: We are planning experiments in other domains, such as bioinformatics and materials science. These require extensive time and domain-specific validation, but we are collaborating with industry partners to conduct real-world evaluations in these areas.
>
> Thus, while the current work focuses on AI, HorizonAI’s architecture and methodology are designed for broader applicability.
>
> ### Weakness 2: Limited Data Sources
> We acknowledge the concern about reliance on arXiv. However:
> - **ArXiv’s Breadth**: ArXiv is a comprehensive repository with broad coverage in fields like physics, mathematics, and engineering.
> - **Adaptability to Other Sources**: HorizonAI can integrate data from sources like Semantic Scholar and Google Scholar. The modular design of PaperTKG ensures seamless incorporation of various data formats.
> While arXiv suffices for this work, the framework is easily adaptable to other sources.
>
> ### Weakness 3: Lack of Expert Review
> We appreciate the concern regarding the absence of expert reviews. Due to time and resource constraints, we were unable to include expert evaluations in this iteration. However:
> - **Mitigation with ResBench Calibration**: We used ResBench, a benchmark calibrated with human-verified data, ensuring LLM scoring aligns with realistic, practical benchmarks.
> - **Transparent and Rigorous Evaluation**: We selected AI topics with well-defined trajectories to cross-validate predictions against recent trends, ensuring reliability.
> - **Scoring Methodology**: Our scoring methodology compares JSON-generated future directions to target subtitles, counting overlaps exceeding a 50% threshold. This ensures objective and consistent evaluation.
> We plan to incorporate expert reviews in future iterations.
>
> ### Weakness 4: Dataset Size
> We understand the concern about dataset size. However, our dataset is carefully curated and effective for the intended evaluations:
> - **Comprehensive Coverage**: We released 20 topics, each with 1 source papers and 10 target papers, resulting in 200 target papers and 20 surveys. Each survey represents the most comprehensive historical summaries for its topic.
>
> - **Large-Scale Knowledge Graphs**: While the visible dataset size may appear small, the underlying knowledge graphs built by HorizonAI are extensive, with each topic incorporating approximately 15,000 papers, totaling around 300,000 papers across all topics.
> - **Focus on Quality**: We prioritized quality over quantity, selecting representative works for each topic to ensure accurate historical synthesis and future predictions.
>
> ## Question: Improvements Over Pre-LLM Methods
> We appreciate your request for clarification on how HorizonAI improves upon pre-LLM methods. Unfortunately, we have not found pre-LLM works addressing the same dual synthesis and prediction task comprehensively. If you have specific references, we would be grateful for the comparison and will include them in future revisions. That said, HorizonAI offers several distinct advantages:
> - **Unified Framework**: Unlike traditional methods that focus on historical synthesis or prediction in isolation, HorizonAI integrates both tasks into a single framework.
> - **Dual-Component Design**: By combining PaperTKG (structured knowledge graphs) with LLM reasoning, HorizonAI enables robust historical modeling and accurate forecasting, leveraging the strengths of both methodologies.
> - **Benchmark Innovation**: ResBench provides a novel evaluation standard specifically for the dual tasks of historical completeness and predictive reliability.
>
> We believe these innovations represent meaningful advancements over existing methods.
>
> ## Conclusion
> We hope this response clarifies the strengths of HorizonAI. By focusing on a defined scope, leveraging a rich dataset, and employing robust methodologies, we demonstrate the framework’s potential. HorizonAI is adaptable to other domains, and we are working to extend its applications to new fields.
>
> We respectfully request that you reconsider your assessment, as we believe the issues raised can be addressed without detracting from the novelty and utility of our approach.
>
> Thank you again for your valuable feedback.
>
> Best regards,
> Authors

---

### Meta-Review · Area_Chair_qfRJ · 2024-12-16

**Metareview:**

After reading the reviewers' comments and reviewing the paper, we regret to recommend rejection.

Taking from reviewer 8usN, this paper proposes HorizonAI, a future research trend forecasting framework inspired by the dual-system theory. In HorizonAI, the Paper2Graph algorithm, which mimics System 1, transforms existing research into temporal knowledge graphs. After that, LLM is leveraged as System 2 for both summarization and prediction through grounded analytical reasoning. The authors collected papers from the arXiv repository, covering 9 distinct topics, and designed a tasked named Dual Temporal Research Analysis. Experimental results on the newly introduced dataset demonstrate that HorizonAI is able to outperform some existing benchmark models, such as AutoSurvey on summarizing historical research and GPT-4o on predicting future developments, respectively.

However, the paper seems to lack relevant citations and comparisons with existing literature, and may lack novelty given the references mentioned in the reviews. There is also a lack of precise details in the evaluation methodology.

We agree with this reviewer's view: the paper seems to lack relevant citations, and falls short in describing the details of the evaluation methodology, that is critical for the approach proposed.

**Additional Comments On Reviewer Discussion:**

The authors have been proactive in addressing the comments raised by the reviewers, and the reviewers were well engaged responding to the authors.

We agree with the reviewers comments, and recommendations, noting that some of the weaknesses may remain and are mentioned in the metareview.

No ethics review raised by the reviewers, and we agree with them.

---

### Decision · Program_Chairs · 2025-01-22

Reject